# Measurement protocols, random-variable-valued measurements, and response process error: Estimation and inference when sample data are not deterministic

**Edward Kroc** * 

Measurement, Evaluation, and Research Methodology Program, Department of Educational and Counselling Psychology, and Special Education, University of British Columbia, Vancouver, Canada

* ed.kroc@ubc.ca

**Data Availability Statement:** The data are simulated for illustration, and reproducible code has been provided in the manuscript's appendix.

## Abstract

Random-variable-valued measurements (RVVMs) are proposed as a new framework for treating measurement processes that generate non-deterministic sample data. They operate by assigning a probability measure to each observed sample instantiation of a global measurement process for some particular random quantity of interest, thus allowing for the explicit quantification of *response process error*. Common methodologies to date treat only measurement processes that generate fixed values for each sample unit, thus generating full (though possibly inaccurate) information on the random quantity of interest. However, many applied research situations in the non-experimental sciences naturally contain response process error, e.g. when psychologists assess patient agreement with various diagnostic survey items or when conservation biologists perform formal assessments to classify species-at-risk. Ignoring the sample-unit-level uncertainty of response process error in such measurement processes can greatly compromise the quality of resulting inferences. In this paper, a general theory of RVVMs is proposed to handle response process error, and several applications are considered.

## 1 Introduction

The customary way to empirically study some stochastic phenomenon $Y$, with corresponding probability distribution $\Pr_Y$ defined over some measurable space, is to observe a set of independent sample realizations of the random variable $Y$: $y_1, \ldots, y_N$. In the broadest sense, these data are a collection of sample measurements of $Y$. If these sample observations are subject to measurement error, then one instead observes the sample measurements $y_1^*, \ldots, y_n^*$, where $Y^*$ is related to the random variable of interest $Y$ in some meaningful way. A host of measurement error models have been proposed to handle such situations (see e.g. [1–7]). Regardless of the presence of measurement error, these sample data are then used to derive estimates and make inferences about various properties of $Y$. In a sense that we will make precise (see Section 2.1),

**Funding:** E.K. was partially funded by the UBC - Paragon Research Agreement throughout the completion of this work. there were no additional internal or external funding sources outside of the UBC–Paragon Research Agreement.

**Competing interests:** The author has declared that no competing interests exist.

such measurement schemes generate full information data sets for the target phenomenon $Y$ since $y_1, \ldots, y_N$, or $y_1^*, \ldots, y_n^*$, are all fixed values (i.e. real-valued constants).

However, there are often instances that arise in an applied data collection system when sample measurements do not generate such full information. Instead, we may find ourselves in a situation where only *partial information* about the sample realizations of $Y$ (or of $Y^*$) is generated by the measurement process due to *response process error*. More precisely, our measurement process may generate observations that are themselves new random variables that describe the chance of observing a particular value, or range of values, for $Y$ or $Y^*$ on each sample draw. What this means is that our *sample measurements* may not generate real numbers; instead, each sample instantiation of the measurement process may generate its own random variable (or, more precisely, its own measure; see Section 2), potentially unique to the sample point.

The phenomenon of response process error has long been recognized as a key problem in psychology (e.g. see [8]), but there are in fact many scientific domains where such measurement phenomena occur. In psychometrics, sample subjects are often subjected to a test or survey to register their amount of agreement with a particular test or survey item. A single multiple choice question on a school quiz is the archetypal example. Standard measurement protocol requires a respondent to identify a subset of the item choices to furnish a response. However, both (subjective) strength of agreement with an item and (subjective) confidence in that agreement create uncertainty in any recorded response, and can help distinguish test-takers who record similar answers. It is uncontroversial that a student who *guesses* the correct answer to a question, and the student who *knows* the correct answer are substantively different for assessment purposes. The fact that these two students would generate identical answers (i.e. sample measurements), though likely with very different degrees of *confidence* in their answers, is a hallmark of response process error.

The question of disentangling ordinary sample uncertainty from such response process error has long occupied psychometricians (in addition to how to disentangle both from more typical measurement error). Certain types of item responses theory models (i.e. 3-parameter IRT models; see e.g. [9, 10]) and CUB models (e.g. [11–13]) are two attempts to account for simple response process errors. Crucially, these techniques are model-based, treating response process error as a population-level—rather than sample-unit-specific—pheonomenon, and do not aim to alter the actual sample data themselves. The framework proposed in this paper takes another tack: collect different sample data (i.e. sample measurements) that encode both "best guesses" and confidence in these guesses for each sample subject. Then work directly with these data to estimate quantities of interest about any underlying trait of interest (e.g. mathematics aptitude). In this way, we will be able to handle many more kinds of response process errors, and our inferences will not require proposing any kind of extra *model*; we alter the *sample data* (i.e. *sample measurements*), rather than any hypothesized structure on the phenomenon of interest. Once we have developed the necessary theoretical vocabulary, we will return to this archetypal scenario in Section 4 to precisely illustrate these points.

In conservation biology, the correct classification of species-at-risk is a vital enterprise that allows governments, private companies, NGOs, and researchers to set informed conservation policies and ensure that vulnerable species are afforded the necessary protections to sustain viable populations. The IUCN Red List categories and criteria is the most widely used tool implemented to perform these assessments [14]. At its most basic, experts assess species' vulnerability according to a variety of quantitative criteria, and then arrive at an at-risk classification. Naturally, it is quite common that complete quantitative information is unavailable for all evaluation items for any particular species. For example, the exact number of mature individuals or the exact reproductive success of a population of breeding individuals is rarely

known. At best, sample estimates for these quantities may be available; at worst, nothing at all is known. Even with the benefit of empirical estimates of the pertinent quantities, there is often considerable disagreement among expert assessers as to the quality and relevance of such estimates for the criteria in question. Nevertheless, IUCN guidelines require assessers to arrive at exact classifications for substantive purposes.

To address this conflict, "fuzzy numbers" [15] and consequent "fuzzy statistics" [16] have been utilized to specify a best estimate and a range of plausible values (see e.g. [17, 18]), combining both empirical estimates from the applied literature for many sub-populations, corresponding sampling and/or measurement error of these estimates, and assessor-specific uncertainties of the quality and applicability of these empirical quantities. In many applications (e.g. those of the previous citations), this framework relies on the use of so-called "triangular numbers." These are particular kinds of fuzzy numbers that naturally arise when one considers an arithmetic of confidence intervals (or, more generally, interval estimates of population parameters), rather than the ordinary arithmetic applicable to point estimates. Consequently, decision-making is a much more delicate process than what arises out of the sometimes overly coarse tradition of the "accept/reject" point-null hypothesis testing paradigm. While a useful approach, these triangular number estimates, similar to the CUB and 3-parameter IRT models of psychometrics, are only equipped to deal with particular types of response process error from a modelling perspective because they still treat the *observed sample data* (i.e. sample measurements) as fixed, deterministic real numbers.

In brief, this paper proposes a mathematical apparatus for quantifying response process error broadly construed via the notion of *random-variable-valued measurements* (RVVMs). All sample data in any situation can be viewed as realizations of a *measurement protocol*, a measure-valued mapping that determines the RVVMs. I will define these ideas formally and show how they generalize and recover the classical case where sample data are totally deterministic (Section 2.1); i.e. sample realizations of a real-valued random variable. I make precise how measurement protocols subject to response process error produce *partial information* about a random variable of interest (Section 2.1), define a calibration condition that is desirable for measurement protocols to obey, and discuss how this condition can be validated in practice (Section 2.2). Problems of estimation and inference are then considered from the point of view of a classical Bayes' estimator (Section 3.1) and the basic case of Bernoulli-valued measurements is further developed as an application (Section 3.2). The next section contains examples and applications (Section 4) that illustrate the utility, power, and novelty of RVVMs in real world research scenarios relevant to field ecology and clinical psychology. A short section comparing and contrasting RVVMs to a few other related statistical topics (Section 5) concludes the paper. Two appendices containing mathematical proofs of propositions and R code for examples follow.

## 2 Measurements as random variables

Let $(\Omega, \mathscr{F})$ be a measurable space. We are interested in the general problem of inferring properties about a random variable $Y : \Omega \to \mathbb{R}$ defined on this measurable space. To do so, we draw a random (finite) sample $\mathscr{S} := \{\omega_i : 1 \leq i \leq n\} \subset \Omega$, and then make a measurement $\rho$ for $Y$ on each sample unit $\omega_i$. The following definition characterizes the mathematical structure of this process.

**Definition 2.1** *For a given random variable $Y$ defined on $(\Omega, \mathscr{F})$, we define a* **measurement protocol** *for $Y$ as a measure-valued mapping $\rho$ such that*

$$\rho := \Omega \to \{\mu \in \mathscr{M}(\mathbb{R}, \mathscr{B}(\mathbb{R})) := |\mu| = 1, \ \text{supp}(\mu) \subseteq \text{Range}(Y)\}.$$

Definition 2.1 requires two things: first, that for any sample point $\omega \in \Omega$, the corresponding *sample measurement* $\rho(\omega) = \mu_\omega$ is a Borel probability measure on the real line. This allows one to explicitly capture any uncertainty in the measurement response process itself at the individual level of the sample point. As we will see, such a feature is essential for quantifying the notion of response process uncertainty in a mathematically coherent way.

Second, the restriction that the support of the sample measurement lies within the image of $Y$ is needed to ensure that $\rho$ actually produces meaningful measurements for $Y$, the random variable of inferential interest. A simple example will illustrate this: take $Y \sim \text{Ber}(\theta)$ defined on the Borel sets over the reals. Consequently, $Y(\omega) \in \{0, 1\}$ for any $\omega \in \mathbb{R}$. A measurement $\rho$ for $Y$ must thus assign a probability that the measurement process produces a 0 or a 1 for any sample point $\omega \in \mathbb{R}$. If the measurement process produces only fixed data (i.e. no response process uncertainty), then $\rho(\omega)$ is a point-mass with support concentrated on the recorded outcome of the observed measurement (i.e. 0 or 1). Moreover, if this measurement process is also free of measurement error, then in fact this point-mass is concentrated on $Y(\omega) \in \{0, 1\}$. Alternatively, if response process uncertainty is present (as in the example of the unsure student answering a test question of the Introduction), then $\rho(\omega) = \text{Ber}(\theta_\omega)$, where the parameter $\theta_\omega$ captures the sample unit (response) uncertainty (i.e. subject-specific confidence) in the correctness of the chosen multiple choice option for the test item in question. If we did not require $\text{supp}(\mu) \subseteq \text{Range}(Y)$, then we could allow a measurement process to assign a non-zero probability to an outcome that could not be produced by $Y$ itself. Such a measurement protocol would not then reflect anything inherently meaningful about the random variable of interest.

## 2.1 Generalized likelihoods

Now we need to make sense of how to use the measurements generated by $\rho$ on our sample $\mathscr{S}$ to actually study the random variable of interest, $Y$. In this paper, we will be concerned with making inferences on global properties (parameters) of the random variable $Y$; e.g. inferring its mean. This is often accomplished probabilistically by creating a likelihood function for the parameter of interest and the observed data. In the classic case, we often write $f(\mathbf{y} \mid \theta)$, where $\mathbf{y} = \{Y(\omega_1), \ldots, Y(\omega_n)\}$, the sample realizations of the random variable $Y$.

Our approach will be no different, except that our measurements are not necessarily simply the sample realizations of $Y$ (or of any measurement error prone proxy for $Y$). Our data, quite literally, consist of the measurements $\rho(\mathscr{S})$, and for each $\omega \in \mathscr{S}$, what we actually observe (via the measurement process) is a measure $\mu_\omega$. In turn, the measure $\mu_\omega$ gives the response certainty that $Y$ assumes some set of values for a given $\omega$. Define random variables $Z_\omega \sim \mu_\omega$. Thus, we write a generalized likelihood as

$$f_Y(\rho(\mathscr{S}) \mid \theta) = f_Y(\mathbf{Z}_\mathscr{S} \mid \theta), \tag{1}$$

where $\mathbf{Z}_\mathscr{S}$ denotes the vector of random variables $\{Z_1, \ldots, Z_n\}$, with $Z_i = Z_{\omega_i}$. This is not recognizable as a true likelihood function since the input $\mathbf{Z}_\mathscr{S}$ is a random variable defined on the product space $\otimes_{i=1}^n (\mathbb{R}, \mathscr{B}(\mathbb{R}))$. However, for every $\mathbf{z}$ in the product space $\mathbb{R}^n$, the vector $\mathbf{Z}_\mathscr{S}(\mathbf{z})$ is a fixed sequence of real numbers (in the range of $Y$); thus, $f_Y(\mathbf{Z}_\mathscr{S}(\mathbf{z}) \mid \theta)$ is an honest likelihood function.

The key observation is that we can now express our generalized likelihood as an average of traditional likelihoods via total probability:

$$f_Y(\mathbf{Z}_\mathscr{S} \mid \theta) = \int_{\mathbb{R}^n} f_Y(\mathbf{Z}_\mathscr{S}(\mathbf{z}) \mid \theta) \; d\boldsymbol{\mu}_\mathscr{S}(\mathbf{z}),$$

where $\boldsymbol{\mu}_\mathscr{S}$ denotes the product measure on $\otimes_{i=1}^n (\mathbb{R}, \mathscr{B}(\mathbb{R}))$ induced by the sequence of

measures $\{\mu_1, \ldots, \mu_n\}$. In what follows, we will always assume that our sample $\mathscr{S}$ is actually a simple random sample, and that our measurement protocol generates *mutually independent measurements*; i.e. that $\mu_i \perp \mu_j$ for all $i \neq j$. We then have the following canonical expression for the generalized likelihood of a simple random sample generated by independent measurements:

$$f_Y(\rho(\mathscr{S}) \mid \theta) = \int_{\mathbb{R}^n} \prod_{i=1}^n f_Y(Z_i(z_i) \mid \theta) \; d\boldsymbol{\mu}_{\mathscr{S}}(\mathbf{z}), \tag{2}$$

where $\mathbf{z} := (z_1, \ldots, z_n)$ is a vector in $\mathbb{R}^n$. The assumption of an independent measurement protocol allows us to write

$$\mathbf{Z}_{\mathscr{S}}(\mathbf{z}) = (Z_1(z_1), \ldots, Z_n(z_n)),$$

and the assumption of simple random sampling allows for the usual decomposition of the joint likelihood into the product of its marginals. As we will see in Section 3, such a generalized likelihood can be used in essentially the same way as any traditional likelihood for purposes of estimation and inference.

It is easy to see that this generalized likelihood collapses down to a traditional likelihood when response process error is absent from the measurement protocol. To recover this classical fixed measurement approach, we simply require our measurement protocol $\rho$ to assign each sample unit $\omega_i \in \Omega$ the appropriate observed realization of the random variable $Y$ at $\omega_i$. As previously noted, this means that the measurement $\mu_i$ is a point-mass concentrated at $Y(\omega_i)$. In this case, we have $\boldsymbol{\mu}_{\mathscr{S}} = \delta_{y_1} \times \ldots \times \delta_{y_n}$, where $\delta_x$ denotes the point-mass at $x$ and we abbreviate with the usual notation $y_i = Y(\omega_i)$. To see how Eq (2) recovers the classical likelihood associated with these fixed measurements, simply compute

$$
\begin{aligned}
\int_{\mathbb{R}^n} \prod_{i=1}^n f_Y(Z_i(z_i) \mid \theta) \; d\boldsymbol{\mu}_{\mathscr{S}}(\mathbf{z}) &= \int_{\mathbb{R}^n} \prod_{i=1}^n f_Y(Z_i(z_i) \mid \theta) \; d\delta_{y_1}(z_1) \times \cdots \times d\delta_{y_n}(z_n) \\
&= \prod_{i=1}^n \int_{\mathbb{R}} f_Y(Z_i(z_i) \mid \theta) \; d\delta_{y_i}(z_i) \\
&= \prod_{i=1}^n f_Y(y_i \mid \theta).
\end{aligned}
$$

Thus, we have that every measurement is the ordinary observed realization of the random variable $Y$, and the generalized likelihood in Eq (2) recovers the classical likelihood function associated to these fixed data.

Notice that exactly the same argument holds if the measurement protocol $\rho$ produced fixed measurements subject to some kind of *measurement error*. In this case, one would observe the sample realizations of $Y^*$, some proxy for $Y$. Crucially, the concept of measurement error assumes that a *fixed* data point is always observed, $Y^*(\omega)$, so again the sample measures generated by such a measurement protocol must be simple point-masses. Of course, in this context, the generalized likelihood in (2) would reduce to $\prod_{i=1}^n f(y_i^* \mid \theta)$, and so the traditional measurement error appartus would apply unchanged, usually in the form of some model relating the

observed proxy $Y^*$ to $Y$, e.g.

$$f(y^* \mid \theta) = \int f(y^* \mid y, \theta) f(y \mid \theta) \ dy. \tag{3}$$

In what follows, we will refer to measurement protocols that generate only point-masses (i.e. fixed data) as *trivial* RVVMs. Such measurements are free of response process error since each sample measurement is deterministic. Absent any measurement error, we should intuitively expect that a set of sample data generated by such a sequence of fixed measurements to contain more *information* about $Y$ than one generated by a sequence of nontrivial RVVMs (i.e. measurements for $Y$ subject to response process error). Indeed, this intuition can be formalized using the classical notions of Shannon information and relative entropy. It is worthwhile to outline this mathematical interpretation here, as it will allow us to formalize what is meant by the phrase *partial information* within the context of RVVMs, and to better understand the unique kind of uncertainty that RVVMs can encode.

Suppose $\rho_0$ is a fixed measurement protocol for $Y$ such that $\rho_0(\omega) = \delta_{Y(\omega)}$, and let $\rho_1$ be any other nontrivial (i.e. not all point-masses) measurement protocol for $Y$ such that $\mu_\omega(Y(\omega)) \neq 0$. Then, for any $\omega \in \Omega$, consider the Kullback-Leibler divergence from $\rho_1(\omega)$ to $\rho_0(\omega)$:

$$D_{KL}(\rho_0(\omega) \mid \rho_1(\omega)) = \int_{\mathbb{R}} \delta_{Y(\omega)}(y) \cdot \log \left( \frac{\delta_{Y(\omega)}(y)}{\mu_\omega(y)} \right) dy.$$

Since $\rho_0(\omega)$ generates a fixed point-mass measure, the Kullback-Leibler divergence reduces to

$$D_{KL}(\rho_0(\omega) \mid \rho_1(\omega)) = -\log(\mu_\omega(Y(\omega))),$$

which is equal to the (always nonnegative) information content of $\mu_\omega$ at $Y(\omega)$. This fact is a reflection of the principle of maximum entropy, and precisely quantifies the amount of information lost when the nontrivial RVVM $\rho_1$ is used as a measurement protocol for the random variable of interest $Y$ at the sample point $\omega$ instead of simply "observing" $Y$ at $\omega$ itself via $\rho_0$. Extending this to the generalized likelihood associated to a set of sample points, we can see that the generalized likelihood associated to the nontrivial RVVMs is naturally more dispersed.

In the above sense, we can say that any nontrivial RVVMs correspond to *partial information* measurement protocols for $Y$. Likewise, we can refer to fixed measurements (i.e. trivial RVVMs) as *full information* measurement protocols for $Y$. Note that full information does *not* imply that the measurement protocol for $Y$ is accurate (e.g. it may still be subject to traditional measurement error), since we could perform the same relative entropy calculation as above even if $\rho_0$ generates point-masses that do not always agree with the sample values of $Y$. The information content of the measurement protocol is only a meaningful reflection of any response process uncertainty that may be present in the measurement process.

Here, it may be useful to recognize that because response process error dictates that *sample measurements themselves* are random quantities, one cannot fundamentally separate (and so quantify) response process error from more customary sources of uncertainty: sampling error and measurement error. Indeed, since we treat all sample measurements as potentially random, the *sample data* themselves are random. Thus, response process error (when it is present) and sampling error are interlocked, since sampling error is quantifiable only with respect to a given dataset. This is *not* inherently meaningful from a frequentist interpretation of probability, but fits well within the tradition of the Bayesian perspective, where probability is usually construed as a measure of certainty rather than a long-run expected outcome. As we will see in

Section 3 when we take up the task of actually estimating population parameters of interest using sample data generated from nontrivial RVVMs, the Bayesian approach will also be far more mathematically natural.

## 2.2 Calibrated RVVMs

Discerning how response process error and traditional notions of measurement error relate is a bit stickier. A proper treatment of this topic requires more pages than this introduction to the mechanics of RVVMs can reasonably hold, but will certainly appear in a forthcoming work. For our present purposes, it is sufficient to distinguish the concepts as we already implictly have: measurement error occurs at a sample unit $\omega \in \Omega$ when a trivial RVVM $\rho(\omega)$ is not supported on the true value of $Y(\omega)$, whereas response process error is generated by non-trivial RVVMs. Both concepts share the common feature that sample data subject to either kind of error cannot be expected to (fully) agree with the true value of $Y$ for any particular sample measurement. Because of this, our measurements must be *calibrated* somehow (i.e. vary systematically in some way around the corresponding true values of $Y$) if we ever hope to quantify the accuracy of any resulting sample estimators for population features of $Y$. Classically, this necessity gives rise to a host of different calibration conditions, usually phrased in the context of one of many different measurement error models (see Kroc and Zumbo [19] for a detailed summary of additive measurement error models, and see Gustafson [6] for a treatment of multiplicative measurement error). Within the context of response process error, we will be concerned with a similar type of calibration condition. As we will see, this condition will ensure that many sample estimators of interest are well behaved.

Fix a measurement protocol $\rho$. For any given $\omega \in \Omega$, define the set

$$\mathcal{O}_\omega = \{\omega' \in \Omega : \mu_{\omega'} = \mu_\omega\}.$$

That is, $\mathcal{O}_\omega$ contains all the sample points $\omega' \in \Omega$ that map to the same measure as does $\omega$ under the the measurement protocol $\rho$. We will assume that $\mathcal{O}_\omega$ is $(\Omega, \mathcal{F})$-measureable, although this is not necessarily so apriori. We then have the following important definition.

**Definition 2.2** *We say that a measurement protocol $\rho$ is* **calibrated** *to $Y$ if for all $\omega \in \Omega$*

$$\int_\mathbb{R} x \; d\mu_\omega(x) = \mathbb{E}_{(\Omega, \mathcal{F})}(Y \mid \mathcal{O}_\omega).$$

Mathematically, the novelty of this definition lies in the fact that it equates expectations on two different probability spaces. Notice that both quantities in play are functions of $\omega \in \Omega$. The lefthand quantity is simply the expectation of the measure assigned to the sample unit $\omega$ via the measurement protocol $\rho$, whereas the righthand quantity calculates the conditional expectation of $Y$ over the subset $\mathcal{O}_\omega \subseteq \Omega$.

Definition 2.2 captures what we would expect to hold if the measurement protocol $\rho$ generates response process error that is still accurate on average; i.e. if it is given by an expert observer (see below). An important consequence of this kind of calibration is that it allows one to easily construct accurate estimators of certain population parameters of $Y$, the actual random variable of inferential interest. This notion of calibration will be exploited in Section 3 to prove asymptotic unbiasedness and consistency of certain Bayes' estimators (see Propositions 3.1 and 3.2), and we will use it again in Section 4 when we consider certain real world instances of RVVMs in greater detail. To establish the former, we will require the following simple result that a *generalized sample mean* is an unbiased and consistent estimator of the population mean.

**Proposition 2.3** (Weak Law of Large Numbers for calibrated RVVMs) *Let $\mathscr{S} = \{\omega_i : 1 \leq i \leq n\}$ be a simple random sample drawn from $\Omega$, and let $\rho$ be an independent measurement protocol on $\Omega$. Define the sample estimator*

$$\bar{\rho}(\mathscr{S}) = \frac{1}{n}\sum_{i=1}^{n}\int_{\mathbb{R}} x \ d\mu_{\omega_i}(x).$$

*Then if $\rho$ is calibrated to Y, some random variable with finite mean and variance, $\bar{\rho}(\mathscr{S})$ is an unbiased and consistent estimator of $\mathbb{E}(Y)$.*

With Proposition 2.3 in mind (see the proof in S1 Appendix), it is worthwhile to spend some time unpacking the practical meaning of the calibration condition of Definition 2.2. In order to derive accurate inferences, it is crucial that the sample measurements $\rho(\omega)$ be reliable in some sense. The calibration condition above says that the expected value of the sample measurement coincides with the expected value of $Y$ over the set of sample points that generate the same sample measurement.

We can quickly see that traditional fixed measurements that are free of measurement error must always be calibrated. Using the logic of the previous subsection, we know that for such a measurement protocol, we can write $\rho(\mathscr{S}) = \{\delta_{y_1}, \ldots, \delta_{y_n}\}$. Of course, the expectation of any random variable, $X$, distributed according to one of these measures is the appropriate $y_i$. Now, notice that for any $i$, we have $\mathscr{O}_{\omega_i} = Y^{-1}(y_i)$. This is simply because we have required our measurement protocol $\rho$ to always return the appropriate fixed value of $Y$ upon measurement. But now

$$\mathbb{E}(Y \mid \mathscr{O}_{\omega_i}) = \mathbb{E}(Y \mid Y^{-1}(y_i)) = y_i = \int_{\mathbb{R}} x \ d\delta_{y_i}(x),$$

for any $i$, since the random variable $Y$ is always constant on the set $Y^{-1}(y_i)$. Moreover, using the standard formula for total variance (see the proof of Proposition 2.3 in S1 Appendix), Var $(Y \mid Y^{-1}(y_i)) = 0$ for any $i$, so $\mathrm{Var}(\bar{\rho}(\mathscr{S})) = \mathrm{Var}(Y)/n$. Proposition 2.3 thus becomes a simple generalization of the Weak Law of Large Numbers for the traditional sample mean.

To understand the calibration property in context, let us consider an example from field ornithology (we will further develop this example in Section 4): assigning the proper sex to a bird captured in the field. An expert researcher may inspect a single individual, denoted by $\omega$, and assess (i.e. measure) the sex as female with 90% confidence (i.e. response certainty). This assessment is the product of a combination of expert diagnostics, including plumage characteristics, body shape, bill shape and size, etc. Importantly, some of these diagnostics may be subjective. We can imagine that this same researcher might capture another bird with the same or different morphological characteristics, and subsequently assess (i.e. measure) the sex of this new bird as female with 90% confidence (i.e. response certainty) again. Since the RVVMs generated by this measurement protocol must be Bernoulli according to Definition 2.1, all birds that generate this same measure of confidence in female sex assessment form the set $\mathscr{O}_\omega$. This researcher's expert assessment is considered calibrated if, among the individuals in this set $\mathscr{O}_\omega$, 90% of them are actually female. This is precisely what we would expect to happen if the researcher performing the measurements is well-trained and knowledgeable; i.e. expert.

This interpretation suggests several ways that the calibration condition in Definition 2.2 can be validated in practice. A researcher could assess a small set of sample elements that are then independently measured exactly (i.e. without response uncertainty), and have their RVVMs checked against these gold standard measurements, with agreement on average yielding calibration. Or, as is common practice in many small bird-banding operations, researchers

can keep track of their measurements and cross-check them for accuracy if and when the sample unit is resampled in subsequent banding seasons. In some cases, the obstacles to assigning a sure classification can disappear upon resampling, as when young banded birds are recaptured at a later stage in life when sexing characteristics are definitive. In still other cases, a sample RVVM can be recorded and further measurement protocols (partial or full information) may be implemented that eventually yield a definitive classification. Such situations can commonly arise in applied medicine when patients are preliminarily diagnosed with a particular affliction, and are then subjected to follow-up examinations and tests to confirm or deny an initial diagnosis.

Even in the absence of rigorous validation, we may reasonably expect the calibration condition of Definition 2.2 to hold as long as the observer assigning the measurement is sufficiently trained and knowledgeable in the aspects of diagnosis and discrimination particular to the research and measurement setting. And inversely, we should not expect such a condition to hold for less experienced assessers. In fact, we may expect such measurements to exhibit important, structural miscalibration, resulting in systemic biases of inferences. This will often be the case in studies of social and psychological phenomena where sample respondents are untrained in uncertainty assessment (e.g. nontechnical survey respondents), or random variables of interest are loosely defined latent constructs.

## 3 Estimation and modelling with RVVMs

In this section, we will investigate the practical use of RVVMs for estimation and inference. Since data generated via a nontrivial measurement protocol generate a generalized likelihood as in Eq (2), it is not immediately apparent how estimation might proceed in a modelling context. The most straightforward approach to estimation and modelling with RVVMs is the Bayesian one, both in terms of direct interpretability of probability as uncertainty and with regards to analytical tractability.

### 3.1 Classical Bayes' estimator with RVVMs

Consider the problem of estimating some model parameter $\theta$ for $Y$ with sample data generated by an independent measurement protocol $\rho$. The classical Bayes' estimator of $\theta \mid \rho(\mathscr{S})$, i.e. the one minimizing the posterior expected value of the mean squared error, is:

$$
\begin{aligned}
\mathbb{E}(\theta \mid \rho(\mathscr{S})) \quad &= \mathbb{E}_{\boldsymbol{\mu}_{\mathscr{S}}}[\mathbb{E}(\theta \mid \rho(\mathscr{S}), \mathbf{z})] \\
&= \int_{\mathbb{R}^n} \int_{\mathbb{R}} \theta f(\theta \mid \mathbf{Z}(\mathbf{z})) \ d\theta \ d\boldsymbol{\mu}_{\mathscr{S}}(\mathbf{z}) \\
&= \int_{\mathbb{R}} \theta \left[ \int_{\mathbb{R}^n} f(\theta \mid \mathbf{Z}(\mathbf{z})) \ d\boldsymbol{\mu}_{\mathscr{S}}(\mathbf{z}) \right] d\theta.
\end{aligned}
\tag{4}
$$

The last line takes the familiar form of the classical Bayes' estimator with a generalized posterior assuming the role of the classical, fixed data (i.e. full information) posterior. Naturally, one could rewrite this posterior in terms of the corresponding likelihood(s), prior, and normalizing factor(s):

$$
\mathbb{E}(\theta \mid \rho(\mathscr{S})) = \int_{\mathbb{R}} \theta \left[ \int_{\mathbb{R}^n} \frac{1}{N(\mathbf{Z}(\mathbf{z}))} f_Y(\mathbf{Z}(\mathbf{z}) \mid \theta)) \ d\boldsymbol{\mu}_{\mathscr{S}}(\mathbf{z}) \right] \pi_0(\theta) \ d\theta.
\tag{5}
$$

Note that the normalizing factor is now a function of the RVVMs (since it is a function of the sample data).

Eqs (4) and (5) suggest just why the Bayesian approach to estimation with RVVMs is fundamentally more mathematically tractable than an optimization based one, such as maximum likelihood. The Fubini-Tonelli Theorem allows for unfettered exchangeability of the integrals over $\theta$ and the joint (probability) measure $\mu_{\mathscr{S}}$ in the definition of the Bayes' estimator, which greatly simplifies the computational problem. On the other hand, a maximum likelihood approach would require optimization of a function of the generalized likelihood (2), a potentially substantial task.

Notice that Eqs (4) and (5) also imply that if $\pi_0$ is a conjugate prior for the traditional likelihood $f_Y(\mathbf{Z}(\mathbf{z}) \mid \theta)$, then $\pi_0$ is a conjugate prior for the generalized likelihood $f_Y(\mathbf{Z}_{\mathscr{S}} \mid \theta)$, in the sense that the generalized posterior

$$f(\theta \mid \mathbf{Z}_{\mathscr{S}}) = \int_{\mathbb{R}^n} \frac{f_Y(\mathbf{Z}(\mathbf{z}) \mid \theta))\pi_0(\theta)}{N(\mathbf{Z}(\mathbf{z}))} \, d\boldsymbol{\mu}_{\mathscr{S}}(\mathbf{z}) \tag{6}$$

is simply a $\boldsymbol{\mu}_{\mathscr{S}}$-weighted average of traditional posteriors that belong to the same parametric family as $\pi_0$. This property can also greatly aid in computation (e.g. see S2 Appendix), and even allows for analytical expressions of the Bayes' estimator in a variety of classic scenarios.

## 3.2 Bernoulli-valued measurements

The most basic such scenario is the study of a Bernoulli phenomenon $Y \sim \text{Ber}(\theta_0)$. In this case, any measurement protocol $\rho$ for $Y$ can only yield Bernoulli-valued measurements $\rho(\omega)$ for any $\omega \in \Omega$ by Definition 2.1. Thus, if $Z_\omega \sim \mu_\omega$, then we may write $Z_\omega \sim \text{Ber}(\theta_\omega)$, where $\theta_\omega \in [0, 1]$ in general, and $\theta_\omega \in \{0, 1\}$ corresponds to a full information RVVM (i.e. no response process error). Given a random sample $\mathscr{S} \subset \Omega$, the generalized likelihood (2) becomes

$$f_Y(\mathbf{Z}_{\mathscr{S}} \mid \theta) = \int_{\mathbb{R}^n} \prod_{i=1}^{n} \theta^{Z_i(z_i)}(1-\theta)^{1-Z_i(z_i)} \, d\boldsymbol{\mu}_{\mathscr{S}}(\mathbf{z}).$$

Since the value of this generalized likelihood is driven only by how many Bernoulli successes occur among the RVVMs, we can simplify this expression by defining the measure $v = \mu_1^* \ldots {}^*\mu_n$ and the random variable $W \sim v$, yielding

$$f_Y(\mathbf{Z}_{\mathscr{S}} \mid \theta) = \int_{\mathbb{R}} \theta^W (1-\theta)^{n-W} \, dv(w).$$

Note that $v$ is not in general a Binomial measure, since the $\mu_i$ measures are not necessarily identical; $v$ is a categorical measure (multinomial on one trial) in general. However, $v$ is discrete on $\mathbb{R}$, so we can simplify things further and write

$$f_Y(\mathbf{Z}_{\mathscr{S}} \mid \theta) = \sum_{k=1}^{n} \theta^k (1-\theta)^{n-k} \text{Pr}_v(W = k).$$

This simplified version of the generalized likelihood can greatly ease the analytical and computational burden of working with Bernoulli-valued measurements, a fact we exploit in the computations of Section 4 (see S2 Appendix for computational details).

Given a prior distribution $\pi_0$ on $\theta$, and using (6), we now have a tractable form for the generalized posterior

$$f(\theta \mid \mathbf{Z}_{\mathscr{S}}) = \pi_0(\theta) \sum_{k=0}^{n} \frac{\theta^k (1-\theta)^{n-k}}{N(k)} \cdot \text{Pr}_v(W = k). \tag{7}$$

Consonant with above, if $\pi_0$ is a generic Beta$(\alpha, \beta)$ prior, then we see that this generalized posterior is simply a $v$-weighted sum of Beta densities. Put another way, conditional on $W = k$, the density is Beta$(\alpha + k, \beta + n{-}k)$.

Using (7) and (4), the Bayes' estimator of $\theta$ becomes

$$
\begin{aligned}
\mathbb{E}(\theta \mid \rho(\mathscr{S})) &= \int_0^1 \theta \cdot \pi_0(\theta) \sum_{k=0}^n \frac{\theta^k (1-\theta)^{n-k}}{N(k)} \cdot \mathrm{Pr}_v(W = k) \ d\theta \\
&= \sum_{k=0}^n \mathrm{Pr}_v(W = k) \int_0^1 \pi_0(\theta) \cdot \theta^{k+1} (1-\theta)^{n-k} \ \frac{d\theta}{N(k)} \\
&= \frac{\alpha + \sum_{k=1}^n k \cdot \mathrm{Pr}_v(W = k)}{\alpha + \beta + n} \\
&= \frac{\alpha + \mathbb{E}_v(W)}{\alpha + \beta + n}
\end{aligned}
\tag{8}
$$

Using a similar argument, we can also derive an analytical expression for the posterior variance:

$$
\mathrm{Var}(\theta \mid \rho(\mathscr{S})) = \mathbb{E}_v[\mathrm{Var}(\theta \mid \rho(\mathscr{S}), W)] + \mathrm{Var}_v[\mathbb{E}(\theta \mid \rho(\mathscr{S}), W)]
$$

$$
= \frac{\alpha(\beta + n) + (\beta + n - \alpha)\mathbb{E}_v(W) + \mathbb{E}_v(W)^2}{(\alpha + \beta + n)^2(\alpha + \beta + n + 1)} + \frac{\mathrm{Var}_v(W)}{(\alpha + \beta + n)(\alpha + \beta + n + 1)}
\tag{9}
$$

In general, the behaviour of this posterior can be quite variable depending on the exact structure of the Bernoulli-valued measurements, even as sample size is increased. In particular, it need not be unbiased or even asymptotically unbiased for $\theta$, and since $\mathrm{Var}_v(W)$ can be on the order of $n^2$ by the sharpness of Popoviciu's Inequality [20], the classical Bayes' estimator need not be consistent for a general independent measurement protocol. However, when we assume that our measurements are *calibrated* in the sense of Definition 2.2, we do have the following interesting result (see the proof in S1 Appendix).

**Proposition 3.1** *Let $\hat{\theta} = \mathbb{E}(\theta \mid \rho(\mathscr{S}))$ be the Bayes' estimator in* (8) *under an arbitrary Beta* $(\alpha, \beta)$ *prior on $\theta$, where $Y \sim Ber(\theta_0)$. Suppose that an independent measurement protocol $\rho$ is calibrated to $Y$ according to Definition 2.2. Then $\hat{\theta}$ is an asymptotically unbiased estimator of $\theta_0$.*

Suppose now that only some of our sample measurements (not necessarily calibrated) generate response process error, while the others are ordinary sample realizations of $Y$ free of measurement error. Then the following proposition holds.

**Proposition 3.2** *Let $\hat{\theta} = \mathbb{E}(\theta \mid \rho(\mathscr{S}))$ be the Bayes' estimator in* (8) *under an arbitrary Beta* $(\alpha, \beta)$ *prior on $\theta$, where $Y \sim Ber(\theta_0)$. Let the first $m_1$ RVVMs generate fixed data (i.e. full information point-masses) free of measurement error, and the remaining $m_2$ be nontrivial RVVMs. Then as $m_1 + m_2 \rightarrow \infty$, $\hat{\theta}$ is an asymptotically unbiased estimator of $\theta_0$ if either*

*(i) $\rho$ is calibrated to $Y$ according to Definition 2.2, or*

*(ii) $m_2 = o(m_1)$; i.e. $\displaystyle\lim_{m_1 \rightarrow \infty} \frac{m_2}{m_1} = 0$.*

*Moreover, if (ii) holds, then $\hat{\theta}$ is a consistent estimator of $\theta_0$; i.e. $\mathrm{Var}(\theta \mid \rho(\mathscr{S})) \rightarrow 0$ if $m_2 = o(m_1)$.*

There are two implicit though important implications of Proposition 3.2 (see the proof in S1 Appendix). The first is that an RVVM mapping need not be calibrated to yield

asymptotically unbiased and consistent estimates. Indeed, condition (ii) is sufficient to ensure such behaviour. Intuitively, this is not surprising given that condition (ii) guarantees that the full and accurate information about $Y$ contained in the subsample of (accurate) fixed measurements will always overpower the partial information contained in the nontrivial RVVMs.

The second important implication is that calibration is not sufficient for consistency of the Bayes' estimator. This too is not surprising when we realize that we have not placed any stabilizing condition on the RVVMs as the sample size grows. Indeed, notice that the first term in the variance expression (9) always approaches zero asymptotically, but that the second term need not. This is a reflection of the fact that the nontrivial RVVMs are theoretically allowed to have as much entropy as we like; thus, we should not expect that their expected value will stabilize as the subsample of nontrivial RVVMs grows. Unless this expectation is stabilized by a relatively greater subsample of fixed measurements, the (generalized) posterior need not converge to a point. Informally, we cannot hope to "sample away" the partial information in our nontrivial RVVMs simply by collecting more of them. In the context of the applied examples considered in the Introduction and in Section 4 below, this phenomenon is totally expected.

## 4 Applied examples and comparisons with methods that ignore response process error

In this section, we illustrate the power and the mechanics of RVVMs to explicitly quantify response process error as part of an integrated data analysis. Results are contrasted with those that would be generated via a typical analysis (i.e. one that ignores response process error) to further emphasize the utility of RVVMs.

### 4.1 Research scenario 1: Sexing birds in the field

Wildlife researchers are often interested in recording the nesting locations of an avian species; however, direct visual confirmation of a nesting site is not always possible [21]. Instead, various diagnostics can be used to assess the likelihood that a nest is present: for example, territorial and mating displays that are characteristic of nesting pairs. Such tentative determinations are an instantiation of the response process error inherent in the measurement process.

In many bird-banding operations individual birds are captured, tagged, and a variety of morphological characteristics are recorded before the birds are released back into the wild. Some measurements are exact, like wing chord and tail length, while others are more diagnostic, like sex and age. In many bird species, sex can be difficult to determine in the field with complete confidence even for highly trained professionals [22]. This is especially true for young birds that have yet to develop their adult plumage [23]. The North American Bird Banding Program requires at least 95% confidence in sex determination before considering a sexing observation valid [24]. Other programs, such as the Vancouver Avian Research Centre, consider at least 90% confidence in sex determination sufficient to generate useable information [25]. Current applied practice dictates that any response process uncertainty is ignored: either when a sample sex identification with 95% certainty is treated as equivalent to a definitive sample sex identification, or when identification uncertainty dips below the acceptable threshold and so the sample datum is discarded. Indeed, this has long been considered good practice in field ornithology for sex and age determination, where Ralph *et al.* [23] have advised us that "it is better to be cautious than inaccurate." Utilizing the framework for handling response process error proposed in this paper, however, we will see that these two options do not have to be exclusive.

Consider the problem of estimating the sex distribution of a population of small songbird that nests in a particular valley in the summer months, a Bernoulli phenomenon denoted by $Y$.

Suppose this songbird requires two years to obtain its full adult plumage, after which identification of sex is exact due to pronounced sexual dimorphism. Both adult and juvenile birds return to the valley of interest each summer, and we aim to quantify the distribution of sexes at the beginning of the breeding season. Juveniles can only occasionally be exactly sexed by plumage or other morphological characteristics; usually, only educated guesses can be made instead. Consequently, sex measurements may be subject to response process error.

Suppose that a team of researchers has been banding and collecting data on these birds for many years. Consequently, they are experienced and well-versed in distinguishing sexes accurately at all ages; i.e. suppose that the measurement protocol that quantifies their response process error is calibrated to $Y$ according to Definition 2.2. Note that the assumed accuracy of the researchers in question could have been explictly justified already, perhaps by cross-checking previous tentative sexing diagnoses of banded juveniles against definitive sexing data on recaptured individuals in subsequent breeding seasons.

For our illustration, suppose 50 birds have been sampled at our test location, and data on individual age (0 = juvenile, 1 = adult), weight (in grams), wing chord length (in cm), and sex (0 = male, 1 = female) have been recorded. Measurement protocols on age, weight, and wing chord length generate full and accurate sample information (i.e. generate trivial RVVMs free of measurement error). We will consider three different measurement protocols for sex in this exploration: $\rho_1$, $\rho_2$, and $\rho_3$. The first, $\rho_1$, generates full and accurate sample information (trivial RVVMs free of measurement error) for all 50 individuals. This is an idealized measurement protocol that is not actually realized in the field; i.e. the measurement protocol that always returns the true sex of the sampled individual with total certainty. The second measurement protocol, $\rho_2$, acts the same as $\rho_1$, but is only applied to the subsample of individuals that can be definitively sexed in the field. Any individuals that cannot be definitively sexed in the field are discarded entirely from the sample, which reflects current best practice in field ornithology [24]. The final measurement protocol, $\rho_3$, will consist of a mixture of trivial and nontrivial, but still *calibrated*, RVVMs. For those individuals that can be definitively sexed in the field, $\rho_3(\omega) = \rho_1(\omega) = \rho_2(\omega)$. However, for individuals that cannot be definitively sexed in the field, $\rho_3$ encodes the certainty that the individual is female, as generated by the domain-expert field technician. Table 1 provides a snapshot of the sample data for each of the three measurement protocols (note that the full dataset(s) can be generated using the R script of S2 Appendix). Note that only 8 birds (all juvenile) are not definitively sexed by the bird-banders.

In the absence of any prior information on sex distribution, it may be natural to expect a uniform split between male and female birds at any age. However, suppose in reality there is more of a tendency for juvenile females to return to their birthplace than for juvenile males, a tendency that disappears once the birds reach adulthood due to differential behavioural changes (e.g. greater pressure on males to find and establish new breeding territory, forcing them to disperse earlier from their natal sites). Unknown to the researchers, suppose the true percentage of adult females at the site of interest is 50%, while the true percentage of juvenile females is 75%. Thus, for our particular data, the dearth of definitively sexed juveniles would inevitably confound any derivative inferences on the sex distribution over the entire population, as well as within the juvenile subpopulation only, if we chose to ignore the nontrivial RVVMs by using measurement protocol $\rho_2$.

For this particular dataset, 19 of the 38 sexed adult birds (all definitively sexed) were female, while only 4 out of the 12 juvenile birds were definitively sexed: all these birds were female. Of the remaining 8 partially sexed juveniles, 5 were actually female (unobserved under the realistic measurement protocols of $\rho_2$ and $\rho_3$).

For the three measurement protocols, we will compare how good of a job the resulting posteriors, and their corresponding Bayes' estimators, do at capturing the true sex distribution in

**Table 1. Example data layout and sample data for the example bird-banding measurement protocols.** Note that the Bernoulli-valued measurements for sex give the observed response process certainty that the sample unit is female.

| index | age | weight | wing.chord | sex, $\rho_3$ | sex, $\rho_2$ | sex, $\rho_1$ |
|---|---|---|---|---|---|---|
| 34 | 1 | 52.087 | 11.239 | Ber(0) | 0 | 0 |
| 35 | 1 | 56.623 | 12.379 | Ber(0) | 0 | 0 |
| 36 | 1 | 57.288 | 10.048 | Ber(0) | 0 | 0 |
| 37 | 1 | 68.327 | 8.315 | Ber(0) | 0 | 0 |
| 38 | 1 | 60.219 | 10.613 | Ber(0) | 0 | 0 |
| 39 | 0 | 38.566 | 11.957 | Ber(1) | 1 | 1 |
| 40 | 0 | 21.984 | 9.999 | Ber(1) | 1 | 1 |
| 41 | 0 | 32.770 | 10.194 | Ber(1) | 1 | 1 |
| 42 | 0 | 26.276 | 11.780 | Ber(1) | 1 | 1 |
| 43 | 0 | 14.701 | 11.553 | Ber(0.9) | NA | 1 |
| 44 | 0 | 11.902 | 11.812 | Ber(0.9) | NA | 1 |
| 45 | 0 | 25.015 | 9.737 | Ber(0.8) | NA | 1 |
| 46 | 0 | 23.797 | 12.210 | Ber(0.8) | NA | 1 |
| 47 | 0 | 12.305 | 11.174 | Ber(0.7) | NA | 1 |
| 48 | 0 | 29.453 | 8.261 | Ber(0.4) | NA | 0 |
| 49 | 0 | 33.304 | 7.370 | Ber(0.3) | NA | 0 |
| 50 | 0 | 27.099 | 7.424 | Ber(0.2) | NA | 0 |

the total population, as well as within the subpopulation of juveniles. We will then complicate the problem by incorporating weight and wing chord data, which will be seen to have differential effects on age and sex. Moreoever, we will compare how the RVVM approach, the only one that explicitly accounts for response process error, compares to a missing data approach where the non-definitively sexed individuals from measurement protocol $\rho_2$ have their sexes imputed.

All numerical calculations were performed in R [26]. Multiple imputations were performed using the 'mice' package [27]. Regression models were fit using the 'RStanArm' package to approximate the appropriate posteriors [28]. All R code is available in S2 Appendix of this article.

**4.1.1 Subgroup analysis.** We will start by examining our estimates of the overall proportion of females. Table 2 contains the values of the Bayes' estimators and the standard deviations of the corresponding posterior distributions when we use the data generated from each of the three measurement protocols. All three estimators assume a naive prior of Beta(15,15) for the overall proportion female.

The estimated proportion of female birds is the same between the full fixed dataset (unobserved), $\rho_1$, and the RVVM-generated dataset, $\rho_3$. This is expected since, as we note in the proof of Proposition 3.2, using calibrated RVVMs does not inject any additional bias into the Bayes' estimator than what would already be present under the complete and accurate information measurements. However, the corresponding posterior distribution under the RVVMs is slightly

**Table 2. Bayes' estimates and posterior standard deviations for proportion of female birds.** A naive prior of Beta(15,15) was used for each of the three estimates.

| | meas. protocol $\rho_1$ | meas. protocol $\rho_2$ | meas. protocol $\rho_3$ |
|---|---|---|---|
| | Full fixed bad hbox | Observed fixed bad hbox | RVVM data |
| est. proportion female | 0.5375 | 0.5278 | 0.5375 |
| standard deviation of estimate | 0.0554 | 0.0584 | 0.0572 |

**Table 3. Bayes' estimates and posterior standard deviations for logistic model: Sex ∼ age.** Coefficients are on the log-odds scale. Measurement protocol $\rho_1$ is response-process-error-free. Measurement protocol $\rho_2$ ignores response process error. Measurement protocol $\rho_3$ quantifies response error via RVVMs.

| | meas. protocol $\rho_1$ | | meas. protocol $\rho_2$ | | meas. protocol $\rho_2$ | | meas. protocol $\rho_3$ | |
|---|---|---|---|---|---|---|---|---|
| | Full fixed data (unobserved) | | Observed fixed data no imputation | | Observed fixed data w/ imputation | | RVVM data | |
| | estimate | s.d. | estimate | s.d. | estimate | s.d. | estimate | s.d. |
| coefficient (intercept) | 1.0973 | 0.6650 | 2.6539 | 1.5390 | 2.8637 | 1.1584 | 1.1516 | 0.7103 |
| coefficient age | -1.0967 | 0.7314 | -2.6202 | 1.5383 | -2.8296 | 1.1880 | -1.1409 | 0.7697 |
| odds of female adult | 1.0005 | | 1.0343 | | 1.0347 | | 1.0108 | |
| odds of female juvenile | 2.9959 | | 14.2098 | | 17.5256 | | 3.1633 | |

more dispersed, reflecting the inherent uncertainty in the nontrivial RVVMs and their use of partial, rather than complete, sample information. Note that the estimated proportion derived from the observed fixed measurements, $\rho_2$, is not as accurate as the RVVM-derived estimate due to the decreased sample size (42 vs. 50) and the lack of partial information use.

Note also that missing data techniques cannot be applied to the data generated by $\rho_2$, simply because we are not utilizing information on any covariates. The RVVM framework of course makes no such requirement.

Now consider what happens if we estimate the proportion of female birds according to age categorization. Here, we will be in a situation amenable to imputation of missing values under measurement protocol $\rho_2$. Table 3 contains the output of logistic regressions for each of our four estimates of interest. Each of the four estimates are derived by assuming a default $N(0, 2.5)$ prior on the 'age' effect and a default $N(0, 10)$ prior on the model intercept.

The RVVM-based model does a much better job than either of the $\rho_2$-generated model fits of reflecting the ideal model fit under the full fixed (unobserved) dataset generated by $\rho_1$. Estimated model coefficients are far more accurate in the RVVM-based fit, and the corresponding posterior standard deviations are naturally wider than those from the idealized $\rho_1$-generated dataset. Again, this reflects the inherent response process error inherent in the measurement protocol $\rho_3$, captured by the nontrivial RVVMs.

Interestingly, the posterior uncertainties under the imputation-based approach are larger than the RVVM-based uncertainties. The reason for this becomes plain when we consider the estimated odds ratios within each age group (bottom two rows of Table 3). Recall that all the definitively sexed juveniles were female; thus, there is no way for an imputation procedure to assign a reasonable chance of observing a juve-nile male, as there are no complete observations on this subpopulation. Such structural confounding yields a severely inflated odds of female sex among juveniles and also inflates the variance in the posterior distributions of the model coefficients.

**4.1.2 Multiple regression with RVVMs.**   We now consider what happens when we fit slightly more complicated regression models in an attempt to uncover finer relationships between the four observed variables: sex, age, weight, and wing chord length. First, we aim to model sex as a function of age and weight. Consider the boxplots in Fig 1. The weight data have been generated so that adult female weights are distributed as $N(50, 5)$ and adult male weights are distributed as $N(60, 5)$ (see the S2 Appendix for reproducible code). For juvenile weights however, the distributions are normal mixtures; this introduces confounding via the measurement process. The idea is that underweight juveniles may be more difficult to defini-tively sex; thus, definitively sexed juvenile females have weights distributed as $N(30, 5)$, while partially sexed juvenile females have weights distributed as $N(20, 5)$. Similarly, definitively sexed juvenile males have weights distributed as $N(40, 5)$, while partially sexed juvenile males have weights distributed as $N(30, 5)$. No weight data are assumed missing.

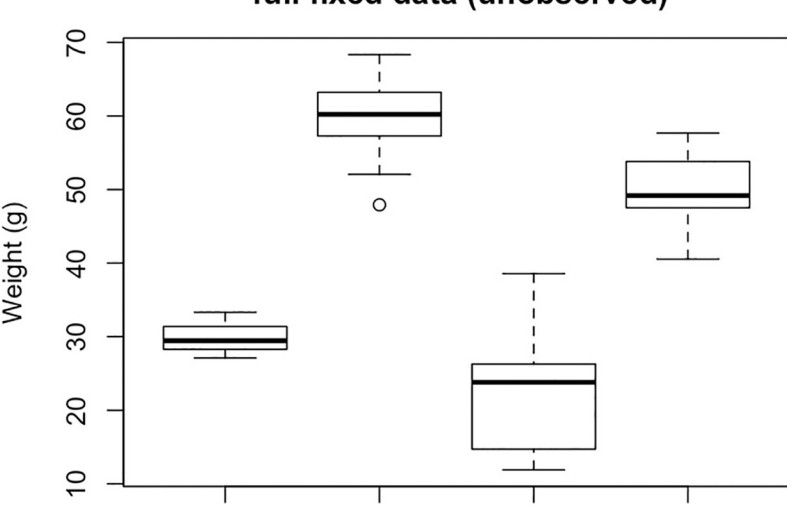

**Fig 1. Weight vs. age and sex for the 50 sampled individuals (not all observed without response process error).**

This type of measurement process confounding has two main effects: (1) all underweight birds will tend to be categorized as female by a missing data approach, and (2) since our data contain no definitively sexed juvenile males, the extra weight covariate will give us no informational leverage with which to model this subpopulation using a missing data approach. In contrast, the RVVM framework will allow us to fix both issues, since calibrated measurements will negate any confounding introduced by the incompleteness mechanism (on average), at the cost of the additional uncertainty generated by the nontrivial RVVMs.

Table 4 contains the output of logistic regressions under each of our four comparison scenarios. Again, each of the four sets of estimates are derived by assuming default $N(0, 2.5)$ priors on all covariates, and a $N(0, 10)$ prior on the model intercept.

Examining the estimated coefficients only, it is quite clear that the RVVM approach closely aligns with the model estimates we would expect if the full true (unobserved) data were

**Table 4. Bayes' estimates and posterior standard deviations for logistic model: Sex $\sim$ age + weight + age $^*$ weight under the three measurement protocols. Coefficients are on the log-odds scale.**

|  | meas. protocol $\rho_1$ | | meas. protocol $\rho_2$ | | meas. protocol $\rho_2$ | | meas. protocol $\rho_3$ | |
|---|---|---|---|---|---|---|---|---|
|  | Full fixed data (unobserved) | | Observed fixed data no imputation | | Observed fixed data w/ imputation | | RVVM data | |
|  | estimate | s.d. | estimate | s.d. | estimate | s.d. | estimate | s.d. |
| (intercept) | 8.0279 | 2.5451 | 20.2487 | 5.9180 | 17.7728 | 5.1105 | 6.5375 | 2.4774 |
| age | 3.1017 | 2.0042 | -0.1926 | 2.4711 | -0.1239 | 2.3734 | 3.4244 | 2.0123 |
| weight | -0.2314 | 0.0815 | -0.2633 | 0.1422 | -0.2324 | 0.1070 | -0.1725 | 0.0785 |
| age $^*$ weight | 0.0263 | 0.0582 | -0.1033 | 0.1155 | -0.0912 | 0.0853 | -0.0113 | 0.0583 |
| odds of female, juv. w/weight = 25 | 9.4199 | | 860,360 | | 156,890 | | 9.2591 | |
| odds of female, juv. w/weight = 30 | 2.9618 | | 230,588 | | 49,089 | | 3.9089 | |
| odds of female, ad. w/weight = 50 | 2.3997 | | 5.6190 | | 4.3392 | | 2.1717 | |
| odds of female, ad. w/weight = 60 | 0.3087 | | 0.1437 | | 0.1706 | | 0.3458 | |

available. Posterior uncertainty in the RVVM framework is comparable to that produced by the fit with the full true (unobserved) data, reflective of the fact that for these sample data, there is very little response process error present. In contrast, the missing data approach performs very poorly.

Examining the estimated raw odds, the full information data generated under $\rho_1$ produce an odds of female among juveniles with weight = 30 g of 2.96, and an odds of female among juveniles with weight = 25 g of 9.42. These estimates are expected when one considers the distribution of the full (unobservd) data over subgroups, as in Fig 1. The estimated odds from the RVVM dataset are similar to the estimated odds from the full $\rho_1$ dataset. In contrast the estimated odds from the $\rho_2$-generated dataset, with or without imputation, are horrendous over the juvenile subgroups. This is not surprising given how the sex data are not missing at random and that there are no definitively sexed juvenile males in our sample. It is important to recognize that the RVVM framework is not susceptible to this same source of confounding (on average) under calibration of the RVVMs. Put another way, there is no need for the partial data to be "incomplete at random," so long as the RVVMs are calibrated.

Next, we aim to model sex as a function of age and wing chord length. Fig 2 displays boxplots for the full information (unobserved) sample data generated under $\rho_1$. Wing chord length was generated as random draws from a $N(11, 1)$ distribution for juvenile females, and adults of both sexes. Thus, missing data in the 'sex' variable are now "missing at random" (MAR).

Wing chord length for juvenile males was generated from a $N(8, 1)$ distribution. Here, we model sex as a function of age and wing chord length, using the same default priors that were used in the previous example. Table 5 displays the results.

Even though the 'sex' data are MAR, the results are similar to the previous example. The model estimates and the resulting estimated odds are still quite bad for the juvenile subgroups with the imputed dataset precisely because even though the missing 'sex' values are MAR, we do not actually observe any definitively sexed juvenile males. The juvenile male group is the only one that generates differential wing chord lengths on average; thus, an imputation procedure will tend to assign these different (smaller) wing chord lengths to the juvenile male

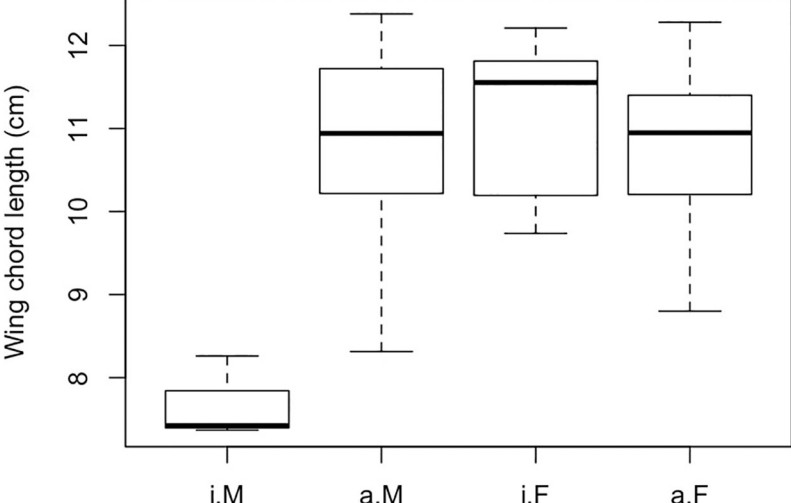

**Fig 2. Wing chord length vs. age and sex for the 50 sampled individuals (not all observed without response process error).**

**Table 5. Bayes' estimates and posterior standard deviations for logistic model: Sex $\sim$ age + wc + age $^*$ wc under the three measurement protocols.** Coefficients are on the log-odds scale. For measurement protocol $\rho_2$, some 'sex' data are missing at random.

| | meas. protocol $\rho_1$ | | meas. protocol $\rho_2$ | | meas. protocol $\rho_2$ | | meas. protocol $\rho_3$ | |
| --- | --- | --- | --- | --- | --- | --- | --- | --- |
| | Full fixed data (unobserved) | | Observed fixed data no imputation | | Observed fixed data w/ imputation | | RVVM data | |
| | estimate | s.d. | estimate | s.d. | estimate | s.d. | estimate | s.d. |
| (intercept) | -6.1633 | 3.1448 | 1.1721 | 4.2750 | -0.1690 | 3.3998 | -3.1373 | 3.1234 |
| age | 1.6367 | 2.0856 | -0.6101 | 2.4232 | -0.1417 | 2.1910 | 0.8495 | 2.1189 |
| wc | 0.7690 | 0.3286 | 0.5625 | 0.5595 | 0.2541 | 0.3403 | 0.4610 | 0.3224 |
| age $^*$ wc | -0.3519 | 0.2173 | -0.5847 | 0.4649 | -0.2353 | 0.2315 | -0.2524 | 0.2214 |
| odds of female, juv. w/wc = 8 | 0.9889 | | 290.65 | | 6.4482 | | 1.7349 | |
| odds of female, juv. w/wc = 11 | 9.9330 | | 1571.2 | | 13.8198 | | 6.9172 | |
| odds of female, ad. w/wc = 11 | 1.0636 | | 1.3744 | | 0.9013 | | 1.0075 | |

group. However, this is a passive prediction born of a lack of alternatives rather than an informed categorization.

In contrast, the RVVM-derived estimates recover the true relationships between the variables, and the estimated odds align well with the estimates on the full and accurate information (unobserved) dataset. Also, note the comparable posterior uncertainties to those generated by the full information (unobserved) dataset; again, a reflection of the relatively small amount of response process error captured by $\rho_3$ vs. $\rho_1$.

Finally, we consider an ordinary normal model for wing chord length as a function of sex and age. Table 6 summarizes the output from these model fits using the same default priors as the previous example.

The RVVM fits are once again closer to the fits given the full and accurate information (unobserved) data, but interestingly, the model fit on the $\rho_2$-generated data with imputation does not actually appear to be too much worse if one only considers the fitted values in each subgroup. However, when attention turns to the model coefficients, it becomes clear why the model fit under measurement protocol $\rho_3$ is preferable to the ones fit under measurement protocol $\rho_2$. Once again, the RVVM-based data produces more accurate estimates since it actually contains partial information on juvenile males, whereas the $\rho_2$ dataset contains no observed complete information on juvenile males. Moreover, there is clear evidence of an average difference in wing chord length between the juvenile male subgroup and any other subgroup when considering either the full information (unobserved) dataset under $\rho_1$ or the RVVM-generated

**Table 6. Bayes' estimates and posterior standard deviations for linear model: wc $\sim$ sex + age + sex $^*$ age for the three measurement protocols.** For measurement protocol 2, some 'sex' data are missing at random.

| | meas. protocol $\rho_1$ | | meas. protocol $\rho_2$ | | meas. protocol $\rho_2$ | | meas. protocol $\rho_3$ | |
| --- | --- | --- | --- | --- | --- | --- | --- | --- |
| | Full fixed data (unobserved) | | Observed fixed data no imputation | | Observed fixed data w/ imputation | | RVVM data | |
| | estimate | s.d. | estimate | s.d. | estimate | s.d. | estimate | s.d. |
| (intercept) | 8.0355 | 0.5817 | 10.9282 | 1.5477 | 9.9403 | 0.9425 | 8.8970 | 0.6954 |
| sex | 3.0617 | 0.6598 | 0.0440 | 1.5143 | 0.5327 | 0.9925 | 1.8773 | 0.7792 |
| age | 2.6759 | 0.6195 | -0.1856 | 1.5348 | 0.7961 | 0.9732 | 1.8165 | 0.7359 |
| sex $^*$ age | -3.0354 | 0.7373 | -0.0906 | 1.5086 | -0.5619 | 1.0592 | -1.8798 | 0.8536 |
| avg. wc length, juv. female | 11.0972 | | 10.9722 | | 10.4730 | | 10.7743 | |
| avg. wc length, juv. male | 8.0355 | | 10.9282 | | 9.9403 | | 8.8970 | |
| avg. wc length, ad. female | 10.7378 | | 10.6961 | | 10.7072 | | 10.7110 | |
| avg. wc length, ad. male | 10.7114 | | 10.7427 | | 10.7364 | | 10.7135 | |

dataset. The corresponding estimated model coefficients clearly separate the mean response in this subgroup from the remainder. Evidence for this separation is substantially weaker under either of the analyses that use the $\rho_2$-generated dataset.

## 4.2 Research scenario 2: Diagnostic rating scales in clinical practice

For our second detailed application, consider a typical scenario in applied psychology where a trained psychologist must diagnose a patient for depression. While many diagnostic techniques and paradigms exist (e.g. see [29, 30]), the Hamilton Depression Rating Scale (HAM-D), or one of its many variants, is a highly classical tool that is still widely used today (e.g. see [31–33]) to aide in the diagnostic process. The HAM-D is a 17-item questionnaire designed to be administered by a health care professional to rate the severity of depression in a patient. The health care professional chooses a single answer for each of the 17 items; item responses are categorical (on 3 or 5 categories), though loosely ordinal in nature, and are assigned integer scores from 0 to 4 (or 0 to 2 for items with 3 categories). Upon completion, the sum of the patient's scores provides a tentative assessment of their severity of depression via the following recommendations (e.g. see [34]):

1. Scores 0—7 = Normal

2. Scores 8—13 = Mild Depression

3. Scores 14—18 = Moderate Depression

4. Scores 19—22 = Severe Depression

5. Scores $\geq$ 23 = Very Severe Depression

Nearly all items of the HAM-D have a clearly subjective component, and this can produce a wide range of response process error as the health care professional attempts to answer each item. Consider item 1 of the scale, aimed at assessing a patient's depressed mood:

1. HAM-D, item 1: Depressed Mood (Gloomy attitude, pessimism about the future, feeling of sadness, tendency to weep)

0 = Absent

1 = Sadness, etc.

2 = Occasional weeping

3 = Frequent weeping

4 = Extreme symptoms

Clearly, there is no objective or consistent distinction between, say, the categories of "occasional weeping" and "frequent weeping." A large amount of subjective interpretation of those terms, as well as how well they apply to the particular patient in question, will inevitably factor into the health care professional's score assignment; i.e. into the sample measurement process. Notice too that this subjectivity can be unique to both the patient being scored and the assessor conducting the scoring. Various construals of semantical uncertainty, blurry categorization, partial ordering, and contextual applicability are all potential instances (see e.g. [17, 35–37]) of response process error.

The traditional approach requires the assessor to simply assign the best-fitting or most appropriate category to the patient for the item: an integer between 0 and 4. An RVVM approach however could require the assessor to indicate their confidence in the applicability of each of the

**Table 7. An encoding of HAM-D sample measurements for 8 patients and two different measurement protocols.**

| Patient ID | $\rho_1$ | $\rho_2$ |
|:---:|:---:|:---:|
| 1 | 1 | (0, 0.5, 0.5, 0, 0) |
| 2 | 1 | (0, 1, 0, 0, 0) |
| 3 | 2 | (0, 0.1, 0.6, 0.3, 0) |
| 4 | 2 | (0, 0.1, 0.8, 0.1, 0) |
| 5 | 1 | (0, 0.7, 0.3, 0, 0) |
| 6 | 3 | (0, 0, 0, 0.8, 0.2) |
| 7 | 2 | (0, 0.1, 0.9, 0, 0) |
| 8 | 1 | (0, 0.6, 0.4, 0, 0) |

5 categories to the particular patient in question. Table 7 summarizes the type of sample data that would be generated by these two different measurement protocols (denoted by $\rho_1$ and $\rho_2$ respectively) for 8 hypothetical patients, assumed to all be assessed by the same health care professional. We have abused notation slightly here and recorded simply the vector parameter that characterizes each (sample) measurement protocol. So for the trivial RVVMs generated by $\rho_1$ corresponding to current practice, we record the support of the point-mass rather than the measure itself. The nontrivial RVVMs generated by $\rho_2$ are always (discrete) categorical measures on 5 categories (the integers 0 to 4); thus, we have simply encoded the measure of these atoms.

Notice that there are considerable differences among, say, all patients who received a score of "1" under measurement protocol $\rho_1$ that are completely hidden by the classical measurement apparatus. Classically, these patients are simply assigned their (subjective) "best" score as decided by their common assessor; thus, they are indistinguishable on this item; i.e. they all receive the same measurement under $\rho_1$. But the use of (nontrivial) RVVMs, $\rho_2$, reveals potentially important clinical differences: e.g. while the assessor is very confident in their score for patient 2, they instead retain considerable uncertainty in their score assignment for patient 1, leaning towards a more severe score. Put another way, a fixed score of "1" seems to mean something quite different for these two patients.

The same two measurement protocols could be applied to every item of the HAM-D, and so opportunities for further clinical distinctions to simultaneously manifest in the sample values of $\rho_2$, and be hidden in the sample values of $\rho_1$, will only accumulate. This means that two different diagnoses for the same patient can be reached depending on which measurement protocol is used. Table 8 summarizes what could happen for our 8 hypothetical patients. Notice that the HAM-D scores agree over the two measurement protocols quite well for some patients, e.g. patient 8, but differ in clinically significant ways for other patients. In particular, patient 6 and patient 7 would both be classified with "Moderate Depression" according to the classical, point-mass measurement protocol, $\rho_1$. However, when accounting for various response process uncertainty in assigning item scores under $\rho_2$, there is a noticeable separation of scores between the two patients. Moreover, under $\rho_2$, patient 6 would be classified with "Severe Depression", while patient 7 would retain the "Moderate Depression" diagnosis.

The sum scores for patients assessed under measurement protocol $\rho_2$ were constructed by using the implied estimator from Proposition 2.3. Recall that this proposition asserted a Weak Law of Large Numbers for calibrated RVVMs, implying that the natural analogue of the traditional sample mean for nontrivial and calibrated RVVMs takes the form

$$\bar{\rho}(\mathscr{S}) = \frac{1}{n}\sum_{i=1}^{n}\int_{\mathbb{R}} x \; d\mu_{\omega_i}(x).$$

**Table 8. Sum scores on the HAM-D rating scale, with corresponding (tentative) diagnoses, for 8 patients and two different measurement protocols.**

| Patient ID | HAM-D Score, $\rho_1$ | Depression diagnosis, $\rho_1$ | HAM-D Score, $\rho_2$ | Depression diagnosis, $\rho_2$ |
|---|---|---|---|---|
| 1 | 15 | Moderate | 16.5 | Moderate |
| 2 | 4 | Normal | 4.7 | Normal |
| 3 | 27 | Very Severe | 25.2 | Very Severe |
| 4 | 24 | Very Severe | 22.9 | Severe |
| 5 | 15 | Moderate | 16.0 | Moderate |
| 6 | 18 | Moderate | 19.1 | Severe |
| 7 | 18 | Moderate | 17.6 | Moderate |
| 8 | 36 | Very Severe | 35.9 | Very Severe |

From this, we see that the natural analogue of the sample sum is $n \cdot \bar{\rho}(\mathscr{S})$; this is the statistic used to compute the HAM-D scores under measurement protocol $\rho_2$ in Table 8. This estimator would also be a natural choice in this context because presumably, if we were following recommended clinical practice, each patient's item scores would be generated by an expert assessor, the health care professional. Thus, we would expect that the nontrivial RVVMs generated by $\rho_2$ will be *calibrated* to the phenomenon of interest according to Definition 2.2.

Notice that this assumption is not usually unique to the measurement protocol $\rho_2$ in applied practice; indeed, if the classical $\rho_1$-generated HAM-D scores were used for clinical decision-making purposes, it would be implicitly assumed that they too were *calibrated* according to Definition 2.2; i.e. that they were accurately measuring the target phenomenon of interest. Psychometricians will often speak of the *validity* of a rating scale, and while that term has many different and often imprecise meanings (see Zumbo & Hubley [8] for thorough discussion), one key facet that the term usually encapsulates is exactly the idea that the measurement in use is fidelitous to the phenomenon. The notion of calibration introduced in this paper is certainly, at least, a part of that idea.

## 5 Discussion

The specific theory for Bernoulli-valued measurements developed and applied in the previous sections can be generalized in a straightfoward manner to categorical-valued measurements. The general theory of RVVMs of course applies equally well to non-discrete-valued measurements, although the analytical niceties of Section 3.2 become far less obvious. Nevertheless, the RVVM framework provides a coherent means of incorporating the quantification of response process error into any applied data analysis, albeit with the caveat that considerable computational power may be required to obtain useable estimates and make valid inferences.

The general idea of response process error and the specific mathematical machinery to quantify it proposed here share many conceptual features with more traditional ideas in the statistics literature, notably: measurement error, fuzzy statistics, elicitation, and missing data. I have already discussed the relationship between RVVMs and measurement error in the preceding sections. Now, consider the other three domains.

As previously indicated in the Introduction, notions from fuzzy numbers/statistics usually arise in practice via the application of "triangular numbers," e.g. [17, 18]. In their more general formulations however (see e.g. [38, 39]), fuzzy numbers are used to extend a real number to a certain kind of real-valued function, or a random variable to a certain kind of set-valued function. Fuzzy statistics tend to operate then as a means to construct new sample estimators from old ones using the arithmetic of fuzzy number systems, but still assuming that the sample data used to construct constituent estimators are deterministic (see e.g. [16]). The RVVM

framework proposed in this paper operates instead by assigning a probability *measure* directly to each sample instantiation of a measurement process, which allows for the possibility that our sample observations are not simply (fixed) numbers or functions. This idea is similar in spirit to the "fuzzy information" approach developed by Okuda *et al.* [40] and Tanaka *et al.* [41], among others, where one assumes that observed sample data can be fuzzy numbers themselves. It would be interesting to investigate what results from this fuzzy information framework can be translated over to our measure-valued one; future work should focus on this.

The idea of *elicitation* (see e.g. [42–44]) aims to use expert information that does not take the form of a fixed measurement of a sample process to improve inferences about the population process. This use of subjective and imprecise expert information makes elicitation conceptually similar to the RVVM framework, specifically to the case of calibrated measurement protocols. However, the two ideas differ substantially in the type of expert information gathered and in how it is eventually used to inform inference. The elicitation method aims to formally build expert information into an informative prior to improve inference; crucially, elicitation does not use expert information to adjust the actual sample measurements that are used to create a likelihood; i.e. it does not attempt to quantify response error in a sample measurement. Put another way, elicitation uses expert information to better calibrate the assumptions behind an inferential model (via a prior), while calibrated RVVMs use expert information to alter the sample data, and so the inferential model, directly (via the likelihood).

Missing data problems have a long history (see Rubin [45]), and techniques for handling them have enjoyed considerable success in a variety of fields (see e.g. [46–48]). Traditionally, the presence of response process error has sometimes been assumed to generate missing data, as in Ralph *et al.*'s [23] recommendations for indefinite age and sexing determinations in field ornithology (see Section 4.1). However, the phenomenon of response process error is not simply a type of missing data problem.

The structural distinctions are easy to make since the missing data framework assumes that all data are fixed (i.e. deterministic), *even those data that are missing*. One either observes a fixed measurement of a random variable $Y$, or one does not. Typically, when some fixed measurements are missing, one then proceeds to leverage information from complete observations on related random variables (covariates) to predict (i.e. impute) the unobserved values of $Y$. Critically, this process requires fixed measurements on auxiliary random variables to get started. Equally important, this process has nothing to say about response process uncertainty inherent in the (sample) measurement process itself.

Moreover, missing data techniques are model-dependent, whereas response process error is an essential feature of the sample data themselves. Partial information due to response process error is *not* the same thing as a total lack of information due to missingness. Indeed, there is a fundamental difference between a measurement process that, say, generates a partial species identification for an individual (say, 50% certainty between two possible species), and one that generates no information by simply not sampling or measuring the sample individual.

It should be clear now that a variety of distinctions exist between the idea of response process error (quantified via RVVMs) and related concepts of imprecise measurement, like traditional measurement error. It is important to note, however, that these different ideas need not occupy distinct domains in applied practice. In fact, it is entirely plausible that an applied researcher may find herself in a situation where the measurement process generates response process error in addition to actual missing data and traditional measurement error. If previous expert information relative to the study phenomena is also available, elicitation could of course be used to inform the priors. Triangular numbers too could be applied to credibility intervals resulting from any analysis to further inform the decision-making process. RVVMs provide a

structured and mathematically coherent way of incorporating partial information due to response process error into an ordinary statistical analysis.

RVVMs arise naturally in a variety of applied research settings. For the most part though, the partial information that they generate has traditionally either had to be simplified (to the detriment of both accurate estimation and reliable inference), or discarded altogether. The theory developed in this paper is only a first step towards a robust and comprehensive theory of this type of sample data, but I contend that it is time to make explicit use of all the information contained in measurement processes subject to response process error.

## Supporting information

**S1 Appendix. Proofs of propositions.**
(PDF)

**S2 Appendix. R code for worked examples.**
(PDF)

## Acknowledgments

I would like to thank Oscar L. Olvera Astivia, Malabika Pramanik, Bruno D. Zumbo, Paul Gustafson, and Louise K. Blight for helpful discussions and useful feedback that helped shape many of the ideas and examples in this paper.

## Author Contributions

**Conceptualization:** Edward Kroc.

**Formal analysis:** Edward Kroc.

**Investigation:** Edward Kroc.

**Methodology:** Edward Kroc.

**Software:** Edward Kroc.

**Writing – original draft:** Edward Kroc.

**Writing – review & editing:** Edward Kroc.

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
