## [Decision Letter · Decision Letter 0]

16 Jun 2020

PONE-D-20-04423

Measurement protocols, random-variable-valued measurements, and response process error: estimation and inference when sample data are not deterministic

PLOS ONE

Dear Dr. Kroc,

Thank you for submitting your manuscript to PLOS ONE. After careful consideration, we feel that it has merit but does not fully meet PLOS ONE’s publication criteria as it currently stands. Therefore, we invite you to submit a revised version of the manuscript that addresses the points raised during the review process.

As you can notice, both the reviewers raised some concerns about the current version of manuscript. Reviewer 1 suggested to clarify some points, especially those regarding the information-theoretic part of the manuscript (Section 2.4). Additionally, he asked for some implications of the proposed approach, for instance w.r.t. hypothesis testing. Reviewer 2 suggested to clarify a number of points of the manuscript, for example the connections with some references and, especially, he asked for empirical evidence which could strengthen your findings. Overall, they converged on a common position according to which the current version of the manuscript must be improved. I have also carefully read the manuscript and I have found it very interesting. It is true that empirical data, particularly those from social sciences, often emerge in a measurement context where there are many sources of uncertainty (e.g., stochastic components, subjective components, measurement errors). I think that statistical modeling aiming at disentangling those sources of uncertainty can enhance statistical findings and conclusions of many scientific research. However, I also have some concerns about the current version of the manuscript. First of all, I would clarify some sections of the manuscript, for instance by reducing Section 1 (it is too verbose now), by reformulate Section 2 in order to show the relevant parts of the modelization (e.g., Section 2.2 can be included into Section 2.1; Section 2.3 needs to be pruned and better justified). Also, I would postpone proofs in Appendix. Similarly, it is a bit obscure to me how Section 2.5 is related to the rest of the manuscript and I would like to see a stronger integration between this part and the evidence emerging from the application. The sample applies for Section 2.4 (as also Reviewer 1 asked). In the manuscript, you often write that modeling measurement uncertainty is crucial for many disciplines, such as applied statistics and social science. And this is true in principle. However, as far as I understand, your application is devoted to a case study from ecology. I would like to see applications from social science (e.g., psychometrics) as well. Otherwise, if not possible, I think some thoughts and connections with other disciplines should be reduced as they generate expectations in the reader that are not satisfied. This is also in line with Reviewer 2 which asked for clarification about the audience of the paper. Another point, which is also highlighted by Reviewer 2, is that regarding other already existing approaches, such as fuzzy statistics. Indeed, as a matter of fact, your approach is very closed to fuzzy random variables. In Section 1 you have declared that fuzzy statistics "[...] does not lend itself to a useful statistical theory" (p.4). However, this is undeserved since it ignores a long tradition of research about fuzzy random variables and fuzzy measurement theory that attempt at modeling subjective uncertainty and, more generally, non-stochastic components of the sampling process. I think you need to elaborate more on such a point, especially since your approach is quite similar to some findings in that literature (e.g., random-set view on fuzzy sets). Finally, the case study needs to be improved. In particular, it should be highlighted how empirical research can benefit from your methodology. As also Reviewer 2 suggested, Section 4 may also benefit from simulation studies, eventually comparing existing methodologies with your proposal (if possible).

I think the manuscript has pontentials and offers novelties on the problem concerning the quantification of the response process error although at this stage it does not provide a clear and concise description of results obtained from the proposed RVVMs approach. 

We look forward to receiving your revised manuscript.

Kind regards,

Antonio Calcagnì, Ph.D.

Academic Editor

PLOS ONE

Journal Requirements:

2. Please clarify in your abstract what is done in the current study; in this case the use of the passive voice obfuscates this manuscript's contribution.

'E.K. was partially funded by the UBC - Paragon Research Agreement throughout the completion of this work.'

Reviewers' comments:

Reviewer's Responses to Questions

**Comments to the Author**

1. Is the manuscript technically sound, and do the data support the conclusions?

Reviewer #1: Yes

Reviewer #2: No

2. Has the statistical analysis been performed appropriately and rigorously? 

Reviewer #1: Yes

Reviewer #2: No

3. Have the authors made all data underlying the findings in their manuscript fully available?

Reviewer #1: Yes

Reviewer #2: Yes

4. Is the manuscript presented in an intelligible fashion and written in standard English?

Reviewer #1: Yes

Reviewer #2: Yes

5. Review Comments to the Author

Reviewer #1: The author proposed a promising mathematical framework for quantifying response process error by means of random-variable-valued measurements (RVVMs), which allows sample data to be viewed as realizations of a measurement protocol, namely, a measurevalued mapping that determines the RVVMs. The manuscript comes with a sounding technical rigor and some useful empirical examples and applications. The introduction provides an exhaustive background and the main intuitions behind the implication of considering the RVVMs framework for empirical practice.

minor comments:

1) In section 2.4, an information-theoretic account of RVVMs is presented. It states that a set of sample data generated by a sequence of fixed measurements contains more information than one generated by a sequence of nontrivial RVVMs. Since in this context the term "information" has a nontrivial meaning, could you please elaborate a little bit more the sentence?

2) I was wondering which might be the implications of adopting the RVVMs framework for Hypothesis Testing, for instance, in relation to the regression coefficients in the Multiple Regression example.

spell checking:

please check the text carefully, for instance:

page 3

quantiative -> quantitative

page 7

probabilisticly -> probabilistically

page 36

eliciation -> elicitation

Reviewer #2: The paper describes an approach to including (subjective) uncertainty about incomplete measurements in statistical analyses, in the form of probability distributions for the (unobserved) precise measurement. The approach corresponds to a form of Bayesian analysis (e.g. Gelman et al., 2013, Bayesian Data Analysis, 3rd edition), and is also at least mathematically equivalent to some definitions of fuzzy data (e.g. Cattaneo, 2017, The likelihood interpretation as the foundation of fuzzy set theory, International Journal of Approximate Reasoning 90).

It is not clear to me what are the goal and the intended audience of the paper. Some aspects of the suggested approach are presented mathematically, but unfortunately with several errors and imprecisions (e.g. unclear distinction between new definitions and consequences of previous definitions). Some data examples are discussed, but the data are artificial and only single datasets are simulated.

I would suggest to:

. clarify the connections and similarities of the suggested approach with other proposals in the literature (the idea of using the expected likelihood has been considered several times: see also the references above);

. improve the mathematical quality of the description and calibrate the mathematical level to the intended audience;

. give examples with real data and/or consider also the average performance of the methods over repeatedly simulated datasets.

6. PLOS authors have the option to publish the peer review history of their article (what does this mean?). If published, this will include your full peer review and any attached files.

Reviewer #1: No

Reviewer #2: No

---

## [Author Response · Author response to Decision Letter 0]

1 Sep 2020

See attached Cover Letter and Response to Reviewers for a detailed response. Thank you!

---

## [Editor Report · Decision Letter 1]

15 Sep 2020

Measurement protocols, random-variable-valued measurements, and response process error: estimation and inference when sample data are not deterministic

PONE-D-20-04423R1

Dear Dr. Kroc,

We’re pleased to inform you that your manuscript has been judged scientifically suitable for publication and will be formally accepted for publication once it meets all outstanding technical requirements.

Kind regards,

Antonio Calcagnì, Ph.D.

Academic Editor

PLOS ONE

Additional Editor Comments:

The current version of the manuscript is more readable now and readers can easily follow ideas and implications of the author's proposal. Overall, I have found the author's ideas and suggestions very interesting and promising. Although there are some overlaps with other research area (e.g., fuzzy random variables, imprecise probability), I think the paper originally sheds light on important issues concerning measurement and uncertainty from a statistical viewpoint.

Finally, I have a minor comment about the Introduction where the author writes about fuzzy statistics. I think the manuscript would benefit from postponing such a comment in Section 5 where he discusses on the connections of RVMMs to other approaches (feel free to do it). Moreover, correct the typos in Section 4.4 (e.g., "thei applicability", "there is are") before sending the final version of the manuscript.

---

## [Editor Report · Acceptance letter]

21 Sep 2020

PONE-D-20-04423R1

Measurement protocols, random-variable-valued measurements, and response process error: estimation and inference when sample data are not deterministic

Dear Dr. Kroc:

I'm pleased to inform you that your manuscript has been deemed suitable for publication in PLOS ONE. Congratulations! Your manuscript is now with our production department.

Kind regards,

on behalf of

Dr. Antonio Calcagnì

Academic Editor

PLOS ONE